# Glycosylation Modulation Dictates Trafficking and Interaction of SARS-CoV-2 S1 Subunit and ACE2 in Intestinal Epithelial Caco-2 Cells

**DOI:** 10.3390/biom14050537

**Published:** 2024-04-30

**Authors:** Marianne El Khoury, Dalanda Wanes, Maura Lynch-Miller, Abdullah Hoter, Hassan Y. Naim

**Affiliations:** Department of Biochemistry, University of Veterinary Medicine Hannover, 30559 Hannover, Germany; marianne.elkhoury@tiho-hannover.de (M.E.K.); dalanda.wanes@tiho-hannover.de (D.W.); maura.lynch-miller@tiho-hannover.de (M.L.-M.); abdullah.hoter@tiho-hannover.de (A.H.)

**Keywords:** glycosylation, protein–protein interaction, *N*-butyldeoxynojirimycin, protein trafficking, SARS-CoV-2

## Abstract

Severe acute respiratory syndrome coronavirus 2 (SARS-CoV-2) mainly targets the upper respiratory tract. It gains entry by interacting with the host cell receptor angiotensin-converting enzyme 2 (ACE2) via its heavily glycosylated spike glycoprotein. SARS-CoV-2 can also affect the gastrointestinal tract. Given the significant role of glycosylation in the life cycle of proteins and the multisystem target of SARS-CoV-2, the role of glycosylation in the interaction of S1 with ACE2 in Caco-2 cells was investigated after modulation of their glycosylation patterns using *N*-butyldeoxynojirimycin (NB-DNJ) and 1-deoxymannojirimycin (dMM), in addition to mutant CHO cells harboring mutations at different stages of glycosylation. The data show a substantial reduction in the interactions between the altered glycosylation forms of S1 and ACE2 in the presence of NB-DNJ, while varied outcomes resulted from dMM treatment. These results highlight the promising effects of NB-DNJ and its potential use as an off-label drug to treat SARS-CoV-2 infections.

## 1. Introduction

The coronavirus disease 2019 (COVID-19) first detected in Wuhan, China, is caused by the severe acute respiratory syndrome coronavirus 2 (SARS-CoV-2), a positive-sense single-stranded RNA virus that led to a pandemic affecting over 773 million individuals and causing more than 7 million deaths as of December 2023 [1]. Signs and symptoms include fever, cough, dyspnea, and loss of taste and smell with glass-ground opacification as observed on chest X-rays of patients [2]. In addition to respiratory symptoms, extrapulmonary manifestations have been observed in the gastrointestinal system and reported by SARS-CoV-2 patients, with signs and symptoms including abdominal pain, nausea, and diarrhea [3,4].

The entry of the enveloped virus into host cells is mediated by the spike (S) glycoprotein, which undergoes a two-step proteolytic cleavage. The first cleavage into S1 and S2 subunits is mediated by the proprotein convertase furin in the Golgi apparatus [5]. The second cleavage step occurs upon contact with the host cell receptor angiotensin-converting enzyme 2 (ACE2), resulting in the cleavage of the S2 subunit into the S2’ subunit by transmembrane protease serine 2 (TMPRSS2) [6,7].

ACE2, also the receptor for SARS-CoV-1 [8], is a carboxypeptidase involved in the renin-angiotensin system (RAS) which plays an important cardiovascular protective role. It is expressed differently across tissues, mostly in the intestine, cardiovascular tissues, and pancreas [9]. ACE2 is co-expressed with host cell proteases involved in SARS-CoV-2 viral entry, such as TMPRSS2, and other potential receptors, such as dipeptidyl peptidase IV (DPPIV) [10,11,12].

The S glycoprotein of SARS-CoV-2 is a heavily glycosylated trimer with *N*- and *O*- glycosylation sites [13]. Each contains 22 potential *N-*linked glycosylation sites that have different glycan types, the most abundant being of the complex type, followed by hybrid glycans and high mannose glycans [14,15]. Glycosylation is an important posttranslational modification in viruses, which exploits the host cell machinery to acquire glycans essential for proper protein folding, trafficking, and immune evasion [16]. Several studies have highlighted that the modulation of glycosylation in many viruses significantly affects their binding to the host, and subsequently their infectivity of polarized cells [17,18]. In the case of SARS-CoV-2, studies have shown that the modulation of glycosylation affects infectivity and viral replication, in addition to reducing S protein synthesis *in vitro* [19,20]. Many clinically available glycosylation modulators could have the potential to be repurposed for the treatment of SARS-CoV-2 infection. The iminosugar *N*-butyldeoxynojirimycin (NB-DNJ), also known as Miglustat, is an FDA-approved drug used for the treatment of Gaucher disease, a lysosomal storage disease, and is also approved in the European Union (EU), Canada, and Japan for the treatment of Niemann-Pick type C [21]. At the cellular level, NB-DNJ is an *N*-glycosylation modulator targeting α-glucosidase I in the endoplasmic reticulum (ER). This enzyme is involved in trimming terminal glucoses from newly synthesized *N-*linked glycosylated proteins, alongside α-glucosidase II. Additionally, NB-DNJ is an inhibitor of glucosylceramide synthase that is involved in glycosphingolipid biosynthesis (GSL) [22]. Previous studies have demonstrated the inhibitory effect of Miglustat on SARS-CoV-2 infectivity [19]. It acts through disrupting viral protein folding and secretion, highlighting its potential as a repurposed drug for COVID-19 treatment. 1-deoxymannojirimycin (dMM) is another *N-*glycosylation modulator that inhibits α-mannosidase I in the Golgi apparatus and possesses anti-HIV activity [23,24].

Given the significant role of glycosylation in the life cycle of proteins, the role of this posttranslational modification in the interaction of S1 with ACE2 in Caco-2 cells was investigated following the modulation of their glycosylation patterns. For this purpose, the glycosylation inhibitors NB-DNJ and dMM were utilized, in addition to mutant CHO cells harboring mutations at different stages of glycosylation. The data demonstrate a significant reduction in the interactions between the altered glycosylation forms of S1 and ACE2.

## 2. Materials and Methods

### 2.1. Chemicals, Reagents, and Antibodies

Cell culture consumables were purchased from Sarstedt (Nümbrecht, Germany). Dulbecco’s Modified Eagle Medium (DMEM), fetal calf serum (FCS), penicillin/streptomycin, Minimal Eagle Medium Non-Essential Amino Acids (MEM NEAA), l-glutamine trypsin-EDTA, 1-deoxymannojirimycin (dMM), Triton X-100 (TX-100), protein A-Sepharose^®^ (PAS) beads, protease inhibitors, and endo-β-*N*-acetylglucosaminidase H (endo H) were purchased from Sigma-Aldrich (Darmstadt, Germany). Gibco Roswell Park Memorial Institute (RPMI) 1640 Medium and Lipofectamine™ LTX Reagent were obtained from Thermo Fisher Scientific (Waltham, MA, USA). *N*-butyldeoxynojirimycin (NB-DNJ) was purchased from Biozol (Eching, Germany). Western blot reagents were purchased from Carl Roth GmbH (Karlsruhe, Germany). The antibodies used are included in Table 1.

### 2.2. Cell Culture

COS-1 (American Type Culture Collection, Manassas, VA, USA), and Caco-2 cells (DSMZ, Braunschweig, Germany) were cultured in DMEM containing 1 g/L and 4.5 g/L glucose, respectively, supplemented with 10% FCS and 100 U/mL penicillin/streptomycin. Calu-3 cells were kindly provided by Prof. Gisa Gerold (Institute of Biochemistry, University of Veterinary Medicine Hannover, Germany) and maintained in DMEM containing 4.5 g/L glucose, 10% FCS, 2% MEM NEAA, 2 mM l-glutamine, and 100 U/mL penicillin/streptomycin. CHO-K1, Lec2, and Lec8 cells [25,26] were obtained from Dr. Rita Gerardy-Schahn (Institute for Clinical Biochemistry, Hannover Medical School, Germany) with permission from Dr. Pamela Stanley (Department of Cell Biology, Albert Einstein College of Medicine Jack and Pearl Resnick Campus, Bronx, NY, USA). CHO cell lines were maintained in RPMI medium supplemented with 10% FCS and 100 U/mL penicillin/streptomycin. All cell lines were passaged and kept at 37 °C in a humidified incubator containing 5% CO_2_ and 95% air. Caco-2 cells were used as a model of enterocytes, as they differentiate upon confluence into polarized columnar epithelial cells and express the small intestinal phenotype, making them a suitable model for studies on intestinal function under physiological and pathophysiological conditions. These cells were maintained up to 5 days post-confluence, when they were further used in our experiments.

### 2.3. Extraction, Transient Transfection, and Pulldown of S1-Fc in CHO Cell Lines and COS-1

CHO cell lines were transiently transfected with cDNA encoding the S1 subunit (pCG1_sol-SARS-2-S1-Fc), kindly provided by Dr. Markus Hoffmann (Infection Biology Unit, German Primate Center, Göttingen, Germany), using Lipofectamine™ LTX Reagent following the manufacturer’s instructions. COS-1 cells were transiently transfected with S1-encoding plasmids using the diethylaminoethyl (DEAE)-dextran method, as has been previously described [27]. The following day, after transfection, COS-1 cells were treated with NB-DNJ (100 µM) or dMM (1 mM) for 24 h.

Media from both CHO and COS-1 cells were collected 48 h post-transfection and treatment with glycosylation inhibitors. The collected media were then centrifuged at 1000× *g* for 5 min to be used for further analysis or interaction experiments with ACE2. Cells were lysed for 2 h at 4 °C with 1% TX-100 in 10 mM Tris-HCl, pH 7.4 and 150 mM NaCl supplemented with a cocktail of protease inhibitors (denoted TX-100 lysis buffer). Crude cellular lysates were then centrifuged at 17,000× *g* for 20 min at 4 °C to remove cellular debris and nuclei. The postnuclear lysates or media of transfected cells were further mixed with PAS beads overnight at 4 °C. The next day, the PAS beads collected from media or cell lysates were washed twice either with PBS (pH 7.4) or TX-100 lysis buffer, respectively, and used for further analysis.

### 2.4. Binding Experiments of S1 to ACE2

Post-confluent Caco-2 cells (and Calu-3 cells, which were used as an alternative model for ACE2 expression, see Appendix A) were treated with either NB-DNJ (100 µM) or dMM (1 mM). After 48 h, the cells were washed twice with PBS, and media obtained from either COS-1 cells or CHO cell lines expressing S1 proteins was added to the Caco-2 cells for 2 h at 4 °C. Cells were then lysed with TX-100 lysis buffer, and S1-Fc was pulled down using PAS beads. The samples were then further analyzed using Western blotting.

### 2.5. Preparation of Brush Border Membranes

Brush border membranes (BBM) from treated Caco-2 or Calu-3 cells were isolated using the modified divalent cation precipitation method [28,29]. The cells were homogenized in BBM buffer (12 mM Tris-HCl, pH 7.0, 300 mM mannitol) supplemented with a cocktail of protease inhibitors using a 26G needle, which was then followed by sonication (3 times, 10 s each, 15 microns). The homogenates were then centrifuged at 5000× *g* for 15 min at 4 °C to remove cell debris, and the supernatant was treated with 10 mM CaCl_2_ and left rotating for 30 min at 4 °C. The samples were then centrifuged at 5000× *g* for 15 min at 4 °C and the pellet (P1) corresponding to the basolateral and intracellular membranes was resuspended in BBM buffer, while the supernatant was subjected to another centrifugation at 25,000× *g* for 30 min at 4 °C. The supernatant (S) was collected and the pellet (P2) representing the BBM was resuspended in BBM buffer. Bradford protein assay was performed to quantify the protein amount in each fraction. Similar protein amounts of each fraction were further analyzed through Western blotting using anti-ACE2 antibodies.

### 2.6. Endoglycosidase H Treatment

Treated and non-treated COS-1 cells were lysed as mentioned previously. PAS beads were used to pull down S1-Fc from lysates and media. The proteins pulled down from the investigated cells, as well as P2 fractions from Caco-2 and Calu-3 cells were first denatured using 5% SDS and 10% 2-Mercaptoethanol for 1.5 h at 37 °C with continuous shaking followed by treatment with 0.5 M sodium citrate, pH 5.5 and 0.5 µL (2.5 U/mL) endoglycosidase H (endo H) for 1 h while on a shaker at 37 °C [30]. Treated and non-treated samples were further analyzed using SDS-PAGE followed by Western blotting.

### 2.7. SDS-PAGE and Western Blotting

Cell lysates or the proteins pulled down with PAS were mixed with Laemmli loading buffer and DTT (10 mM). The samples were then denatured at 95 °C for 5 min, resolved on 8% gels [31], and blotted on PVDF membranes. The membranes were blocked in 5% skimmed milk in PBS supplemented with 0.1% Tween-20 for 1 h at room temperature before the respective primary and secondary antibodies were added (see Table 1). Typically, the antibody binding to its respective antigen was performed for 1 h in 2% skimmed milk in PBS, 0.1% Tween-20, followed by three washing cycles with the same buffer. Finally, HRP-conjugated secondary anti-mouse or anti-rabbit antibodies were added for another 1 h followed by another three washing cycles in the same buffer as above. The ChemiDoc MP™ Touch Imaging System (Bio-Rad, Munich, Germany) was used to detect chemiluminescence signals. The quantification of the protein band intensities was performed with Image Lab software 6.1.

### 2.8. Statistical Analysis

All experiments were performed at least 3 times and are presented as mean ± standard error of the mean (S.E.M.). A one-way and two-way ANOVA followed by Tukey’s multiple comparisons test or an unpaired t-test were used for result assessment. The statistical analysis was carried out using GraphPad Prism 9.0.0 (GraphPad Software, San Diego, CA, USA). The statistical significance is represented as follows where appropriate * *p* < 0.05, ** *p* < 0.01, *** *p* < 0.001, **** *p* < 0.0001.

## 3. Results

### 3.1. Glycosylation Modulation in Mutant CHO Cells, Lec2 and Lec8, Affects Trafficking and Secretion of S1

Lec2 and Lec8 cells are genetically modified CHO cells, in which the transport of CMP-sialic acid or UDP-galactose, respectively, into the Golgi apparatus is blocked. They have been well documented and biochemically characterized, as has the glycosylation of proteins expressed in these cells [25,26,32,33]. Lec2 and Lec8 therefore provide exquisite platforms to express glycoproteins with specific glycan modifications and to elucidate the implications of glycosylation steps within the medial Golgi and *trans*-Golgi on protein trafficking. As shown in Figure 1, S1 expressed in Lec2 and Lec8 were secreted into the external milieu and revealed slightly smaller apparent molecular weights, compatible with their deficient glycosylation in the Golgi. The secretion of modified S1 into the media of both Lec2 and Lec8 cells was significantly reduced, by 25% and 80%, respectively, as compared to secreted S1 in the control CHO-K1 cells (Figure 1A,B). Importantly, the level of intracellular S1 in Lec2 cells was substantially higher than in its control counterpart, indicative of a trafficking block and intracellular accumulation of this S1 glycoform (Figure 1C,D). In fact, the comparison of the ratio between secreted and intracellular S1 revealed an 81% reduction (Figure 1E). On the other hand, the intracellular levels of S1 expressed in Lec8 cells were slightly lower than those of control S1 (Figure 1C,D), and the ratio between secreted and intracellular S1 showed a marked reduction that was, however, statistically not significant (Figure 1E). Taken together, the reduced expression levels of the modified S1 glycoforms imply that the altered glycosylation of S1 has affected its trafficking and secretion.

### 3.2. Glycosylation Modulation Impacts the Interaction of S1 with ACE2 in Intestinal Epithelial Cells

Next, the interaction of the modified glycoforms of S1 with ACE2 in intestinal Caco-2 cells was investigated. For this purpose, the culture medium collected from the transiently transfected CHO cell lines containing secreted S1 proteins was added to Caco-2 cells for 2 h at 4 °C to allow potential binding while inhibiting endocytosis. Cells were then lysed, and S1 was pulled down using PAS beads that bind its Fc tag, followed by Western blot. Figure 2 shows an almost 4-fold decrease in the interaction between ACE2 and S1 that has been secreted from Lec8 cells as compared to S1 secreted from Lec2 cells and about 2-fold decrease as compared to S1 secreted from wild type CHO-K1 cells. These data suggest that the type of glycosylation of S1 can be decisive in both directions, in increasing or decreasing the binding of S1 to ACE2 and underlines the critical role of glycosylation of the S protein. The terminal sugar in S1 secreted from Lec2 cells is galactose, due to the deficient addition of sialic acid in these cells. This result implies that the increased binding of S1 to ACE2 is likely due to the exposed galactose on S1. In contrast, Lec8 cells are deficient in the galactosylation step and revealed reduced binding of S1 to ACE2.

### 3.3. N-glycosylation Modulators Affect S1 Trafficking and Secretion

Obviously, glycosylation affects the interaction between S1 and ACE2. To corroborate these data using another approach, the effects of glycosylation modulators on the trafficking and secretion of S1 proteins and their binding to ACE2 was investigated. Here, COS-1 cells expressing S1 were treated with the *N*-glycosylation modulators NB-DNJ and dMM for 24 h. As shown in Figure 3, pull-down experiments with PAS beads revealed S1 in the cell culture medium of both treated and non-treated cells, though its secretion was affected to varying extents in the treated cells. In NB-DNJ-treated cells, a significant reduction of approximately 80% in the level of secreted S1 was detected, as compared to non-treated cells (Figure 3A,B). Calnexin and β-actin were utilized as internal controls that are presumably not affected by this treatment. The expression levels of these two proteins were similar in the treated and non-treated cells, indicating that the effects of NB-DNJ on S1 secretion are restricted to this protein or any other proteins along the secretory pathway, but not due to other cellular mechanism, for instance cell death (Appendix A). dMM also affected the secretion of S1, albeit to a lesser extent of 10% (Figure 3C,D). Moreover, the ratio between secreted and intracellular S1 showed an 82% and 44% reduction in secretion for NB-DNJ-treated (Figure 3C) and dMM-treated cells (Figure 3F), respectively.

Both the secreted and intracellular forms of S1 were treated with endo H to verify their glycosylation patterns. This enzyme cleaves high-mannose and hybrid structures present in the ER and along the early secretory pathway (up to the *cis*-Golgi), while mature complex glycosylated proteins in the Golgi are resistant to endo H. Figure 3G shows the intracellular form isolated from the cellular lysates to be endo H-sensitive and shifted to a lower molecular weight, indicating that its glycosylation is of the mannose-rich type typical of the ER. As expected, the S1 protein secreted into the cell culture medium was endo H-resistant, compatible with its trafficking along the secretory pathway and processing to a complex glycosylated form in the Golgi apparatus. Similarly, the S1 proteins isolated from the cellular lysates of the NB-DNJ- and dMM-treated cells also revealed endo H-sensitive protein bands; the secreted S1 proteins from the treated cells, however, varied in their reactivity with endo H. While S1 from NB-DNJ-treated cells was endo H-resistant and therefore complex glycosylated, the S1 from dMM-treated cells retained a predominant endo H-sensitive form due to the inhibition of mannose trimming by α-mannosidase I.

Given that NB-DNJ inhibits glucose trimming by α-glucosidase I in the ER, thus affecting the potential interaction of S1 with calnexin, we propose that the substantial reduction in the proportion of secreted S1 and the increased intracellular S1 levels in the presence of NB-DNJ can be attributed to impaired folding of the protein in the ER.

### 3.4. The Trafficking of ACE2 to the Brush Border Membrane of Caco-2 Cells Is Greatly Affected in the Presence of NB-DNJ and dMM

The trafficking of ACE2 to the apical membrane, or brush border membrane (BBM), in Caco-2 cells was assessed 48 h after the treatment of the cells with the glycosylation modulators NB-DNJ and dMM. Here, the calcium-based fractionation of the cellular homogenates (H) was utilized to separate intracellular and basolateral membranes (P1) from the apical membrane or BBM (P2) and cytosolic vesicles (S). Western blot analysis of the different cellular fractions revealed ACE2 in all fractions, though predominantly in the BBM, of treated and non-treated Caco-2 cells (Figure 4A). Nevertheless, the protein levels of ACE2 in the BBM varied according to the treatment conditions, highlighting the effects of the two inhibitors on the final sorting of ACE2 to the apical membrane. The ratio of ACE2 in P2 versus ACE2 in the total homogenates was (P2/H) 3.34 ± 1.22 in the presence of NB-DNJ and was therefore significantly less than 8.13 ± 1.83 in the control non-treated cells, clearly indicating that the sorting of ACE2 had been markedly impaired. Treatment with dMM led to a reduction in ACE2 levels in the BBM as reflected by a P2/H ratio of 5.81 ± 1.01 (*p*=0.20) (Figure 4A,B).

The glycosylation pattern of ACE2 that had been transported to the BBM was further examined using endo H (Figure 4C). The results show that ACE2 in the NB-DNJ-treated and the control non-treated cells was endo H-resistant, indicating that it is complex glycosylated. On the other hand, dMM-treated ACE2 was endo H-sensitive, as is expected after the inhibition of α-mannosidase I (Figure 4C). Similar results were observed upon subjecting control and treated samples to endo H treatment (Appendix A). Taken together, these results demonstrate that *N*-glycosylation modulators differentially affect the trafficking of ACE2 to the apical membrane.

Since SARS-CoV-2 initially targets the respiratory tract, Calu-3 cells have been utilized in initial phases of the current study as a model of bronchial epithelial cells to establish and confirm the functionality of the experimental set up with intestinal Caco-2. Interestingly, variations in the trafficking of ACE2 to the apical membrane were observed between Calu-3 and Caco-2 upon treatment with NB-DNJ that could be cell type-dependent relevant to the cell morphology, i.e., extent of cell polarity, and final polarized sorting events of ACE2 to the apical or basolateral membranes in these cells (Appendix A). It should be mentioned that intestinal Caco-2 cells exhibit the strictest polarity among epithelial cells with the most pronounced brush border/apical membrane rendering sorting of proteins, such as ACE2, to the apical membrane, an event that requires high-fidelity sorting signals [34].

### 3.5. Altered Interaction of the Modified Glycoforms of S1 with ACE2

Having assessed the effects of glycosylation modulators on the secretion of S1 and the trafficking of ACE2 in Caco-2 cells, the interaction of the modified glycoforms of these two proteins was investigated following procedures similar to those described above (see Section 3.2). As shown in NB-DNJ-treated Caco-2 cells, the interaction between S1 and ACE2 was considerably reduced by 72%, as compared to the non-treated control glycoproteins (Figure 5A,B). This interaction was further reduced by 74.5% when both COS-1 and Caco-2 cells were simultaneously treated with the glucosidase inhibitor. The reduction in interaction observed is due to decreased secretion of S1 and impaired trafficking of ACE2 in the presence of NB-DNJ. No significant variation in binding was observed when only Caco-2 cells were treated with NB-DNJ (Figure 5A,B), presumably due to adequate amounts of apically expressed ACE2 available to interact with secreted S1. The treatment of both cell types with dMM led to a reduction in interaction between the modified glycoforms of S1 and ACE2 (Figure 5C,D). However, the individual modification of S1 with dMM did not impact the interaction of the glycoforms. The treatment of Caco-2 cells alone with dMM led to an increase in interaction between ACE2 and S1 suggesting that non-modified S1 may exhibit a higher affinity towards the high mannose form of ACE2 (Figure 5C,D).

## 4. Discussion

SARS-CoV-2 is at the heart of the pandemic that started in December 2019, and, despite the presence of vaccines, therapeutic strategies are still needed to combat the infection. In addition to respiratory symptoms, many extrapulmonary manifestations have been observed in the gastrointestinal tract and reported by SARS-CoV-2 patients, including abdominal pain, nausea, and diarrhea [3,4]. SARS-CoV-2 interacts with host cell receptors and hijacks cellular machinery. It undergoes glycosylation, allowing the virus to evade the host immune system as these added sugar moieties are recognized as self-antigens [35]. Glycosylation may therefore constitute a potential target for therapeutic measures.

The S glycoprotein has been shown to exhibit lectin-like properties, since it is able to recognize carbohydrates [36]. To highlight the importance of glycosylation for proper trafficking and secretion of the S1 subunit, mutant CHO cell lines were used as *in vitro* models with various deficiencies along the glycosylation pathway [25,26]. Lec2 and Lec8 cells have a defect in CMP-sialic acid transport and UDP-galactose, respectively [25,26]. Galactose is added in the *trans*-Golgi following the trimming of mannoses and the subsequent addition of two N-acetylglucosamine [37]. Sialic acid is then added as a final step before secretion [38]. The lack of these two sugars is noticeable via the shift in the electrophoretic pattern of S1 secreted from Lec2 and Lec8, while intracellular S1 retains its mannose-rich glycosylation typical of the ER and the early secretory pathway. In the case of Lec2 cells, S1 accumulated intracellularly, thereby leading to a lower secretion rate. The lack of sialic acid in Lec2 cells may cause the accumulation of S1 in the Golgi apparatus and subsequent ER stress, as has been demonstrated for glycoconjugates that accumulate in the Golgi apparatus rather than in the plasma membrane [39]. Moreover, the removal of sialic acid and reduced fucosylation have been shown to enhance the reactivity of neutralizing antibodies [40]. Collectively, these findings highlight the importance of sialic acid in the proper secretion of S1 and propose it as a potential therapeutic target.

Repurposing drugs approved for other diseases could potentially be effective and time-saving in treating SARS-CoV-2 infections, as these drugs have already been tested for safe administration to humans [41]. In our study, the impact of the glycosylation inhibitors NB-DNJ and dMM on the interaction between S1 and ACE2 and on their individual behavior along the secretory pathway was investigated. NB-DNJ and dMM are glucose and mannose analogs that bind to the active site of the target enzymes, inhibiting their activities [42,43]. The results with the α-glucosidase inhibitor NB-DNJ show the importance of the first glycosylation steps for the proper secretion of S1, as NB-DNJ significantly reduced the secretion of the S1 protein. ER α-glucosidases trim the glucoses on proteins prior to their interaction with calnexin/calreticulin [44]. Therefore, the inhibition of these ER-located glucosidases leads to an increased retention of glucosylated proteins in the ER, likely due to alterations in protein folding as has been shown previously [19,45,46,47]. These results are concomitant with the effect of other iminosugars that have been investigated. For example, celgosivir and its active form, castanospermine, were found to inhibit the replication of SARS-CoV-2 and to reduce spike protein synthesis in a cell culture model [20]. An inhibitory effect on SARS-CoV-2 infectivity and spike glycoprotein secretion has also been described for Miglustat [19]. Although secretion was significantly reduced, it is obvious that complex glycosylated proteins were still secreted post-treatment as confirmed by endo H treatment. This result suggests that another mechanism might be involved, allowing proteins to continue moving along the secretory pathway. In fact, proteins that escape ER quality control and reach the Golgi are processed by endo-α-1,2-mannosidase, enabling the escaped proteins to bypass the classical secretory pathway [48,49,50,51].

The treatment of intestinal epithelial Caco-2 cells with NB-DNJ led to the reduced trafficking of ACE2 to the BBM and resistance of this receptor to endo H, as was also shown for S1 secretion. As explained previously, other mechanisms post-ER could be involved in the glycosylation pathway, resulting in the maturation and formation of complex glycosylated proteins. In fact, a study by Beimdiek et al. shows that despite a deficiency in glycosidase 1 found in the congenital disorder of glycosylation IIb (CDG-IIb) patients, complex glycosylated proteins can be detected [52]. These findings further confirm the presence of a glucosidase-independent pathway. The interaction between S1 and ACE2 was significantly reduced either when COS-1 cells were treated alone or when COS-1 and Caco-2 cells were treated simultaneously. However, the treatment of Caco-2 cells alone did not affect the interaction. Since S1 secretion is significantly reduced upon the treatment of COS-1 cells, as previously described, both cases of reduced interaction can be attributed to the presence of fewer S1 proteins in the medium, as the exclusive treatment of Caco-2 cells did not affect the interaction. Therefore, the effect that NB-DNJ exclusively exerts on the S1 subunit makes it a potential drug for the treatment of SARS-CoV-2 infection.

Until present, no studies on the effect of the α-mannosidase I inhibitor dMM on SARS-CoV-2 have been documented, despite its antiviral properties having been shown for HIV [53]. In the presence of dMM, S1 is secreted normally, despite incomplete maturation, and trafficking of the dMM-modified glycoform of ACE2 to the brush border membrane also remains unaffected. Nevertheless, the modifications of S1 and ACE2 together negatively impacted the interaction of these glycoforms. In view of the complete endo H-sensitivity of S1 and ACE2 in the presence of dMM, and thus their predominantly mannose-rich glycosylation patterns, alterations of folding determinants in S1 and ACE2 can be postulated to lower their binding affinities. The importance of glycosylation to the binding of the SARS-CoV-2 spike to ACE2 has been demonstrated through the deletion of specific glycosylation sites on the host cell receptor that revealed differential effects on the interaction between the SARS-CoV-2 spike and ACE2 [54]. The exposure of galactose residues on ACE2 through the inhibition of sialic acid terminal glycosylation enhanced the binding of the S glycoprotein to the host cell receptor [55,56].

## 5. Conclusions

In conclusion, our study provides biochemical insights into the importance of glycosylation and the effect of its inhibitors on the trafficking and interaction of the S1 subunit of SARS-CoV-2 with ACE2 in intestinal epithelial cells. The data show that interference with glycan processing significantly impacts the trafficking of S1 and ACE2 and their interaction. As a result, glycosylation inhibitors that are approved for administration in humans could potentially be prescribed off-label to treat SARS-CoV-2 infections, since drug repurposing is a cost-effective and time-saving treatment strategy. In particular, the inhibitory effect of NB-DNJ on S1 secretion, ACE2 trafficking, and subsequently reduced S1/ACE2 interaction provides a promising therapeutic target. The utilization of secreted S1 permits binding studies under suitable, almost native conditions, in which the culture medium containing S1 is added without further purification to the cells. Moreover, S1 comprises the receptor-binding domain (RBD), responsible for the interaction with ACE2. Therefore, utilizing it in studies on the interaction with host cells via ACE2 is an experimental model that partially reflects the *in vivo* situation. Along these lines, further investigation of SARS-CoV-2 replication in intestinal epithelial cells in the presence of these modulators is necessary to investigate whether these effects can be translated into the *in vivo* infection situation.

## Figures and Tables

**Figure 1 biomolecules-14-00537-f001:**
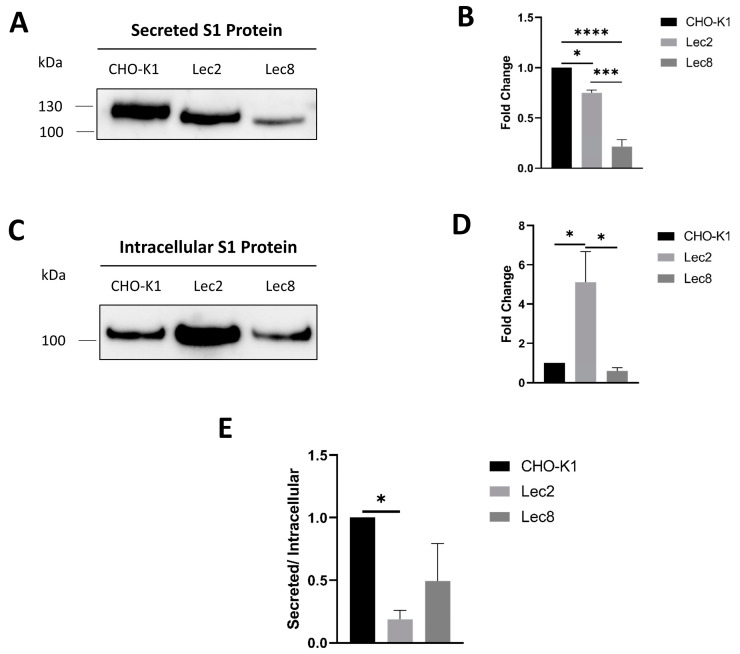
Glycosylation modulation in mutant CHO cells, Lec2 and Lec8, affects trafficking and secretion of S1 (**A**) CHO-K1, Lec2, and Lec8 cells were transiently transfected with S1. The secreted form was then pulled down using PAS beads and analyzed using Western blot. (**B**) Analysis of the results obtained in A. (**C**) The intracellular form was collected after lysis of the transiently transfected CHO cell lines and analyzed using Western blot. (**D**) Analysis of the results obtained in C. (**E**) Ratio of secreted to intracellular S1 proteins. Tukey’s multiple comparisons test, * *p* < 0.05, *** *p* < 0.001, **** *p* < 0.0001, versus CHO-K1, S.E.M., *n* = 3. Original Western blot images are contained in Appendix A.

**Figure 2 biomolecules-14-00537-f002:**
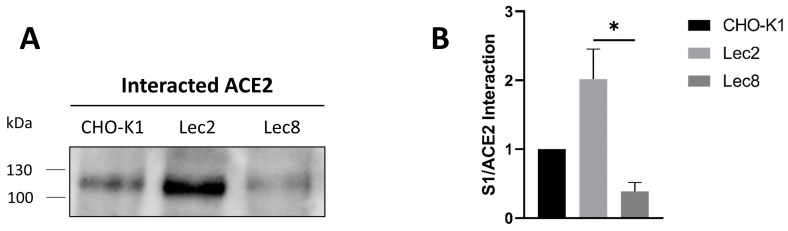
Glycosylation modulation impacts S1/ACE2 interaction in intestinal epithelial cells. (**A**) Culture media containing secreted S1 proteins from CHO cell lines expressing S1 was collected and added to Caco-2 cells for 2 h at 4 °C. Cells were then lysed and S1 was pulled down using PAS beads. The captured S1 proteins were subjected to SDS-PAGE followed by immunoblotting with anti-ACE2 antibodies. (**B**) Analysis of the results obtained in A. Tukey’s multiple comparisons test, * *p* < 0.05, versus CHO-K1, S.E.M., *n* = 3. Original Western blot images are contained in Appendix A.

**Figure 3 biomolecules-14-00537-f003:**
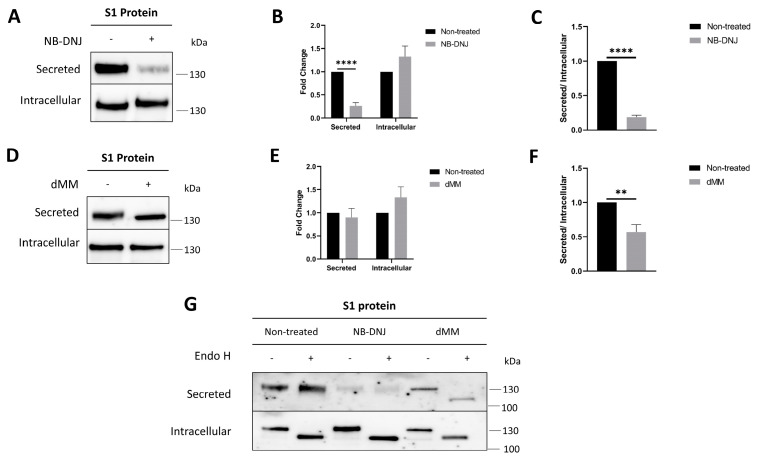
*N*-glycosylation modulators affect S1 trafficking and secretion. (**A**) COS-1 cells expressing S1 were treated with NB-DNJ for 24 h. The secreted and intracellular forms were analyzed using Western blot. (**B**) Analysis of the results obtained in A. (**C**) Ratio of secreted to intracellular S1 proteins in NB-DNJ-treated cells. (**D**) COS-1 cells expressing S1 were treated with dMM for 24 h. The secreted and intracellular forms were analyzed using Western blot. (**E**) Analysis of the results obtained in C. (**F**) Ratio of secreted to intracellular S1 proteins in dMM-treated cells. (**G**) Secreted and intracellular S1 proteins were treated with endo H, followed by Western blot analysis. The values represented are normalized to the non-treated control. Unpaired *t*-test, ** *p* < 0.01, **** *p* < 0.0001 versus non-treated, S.E.M., *n* = 3. Original Western blot images are contained in Appendix A.

**Figure 4 biomolecules-14-00537-f004:**
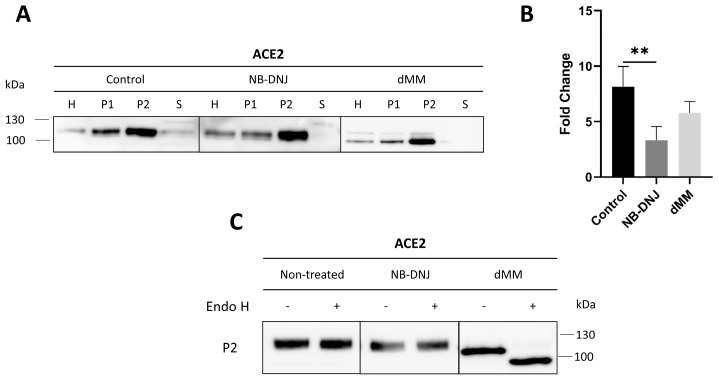
The trafficking of ACE2 to the brush border membrane of Caco-2 cells is greatly affected in the presence of NB-DNJ and dMM. (**A**) Caco-2 cells were treated with NB-DNJ and dMM for 48 h. Total homogenates (H) were fractionated into intracellular and basolateral membranes (P1), apical membrane/BBM fraction (P2), and a fraction containing cytosolic vesicles (S). The fractions were then analyzed using Western blot. (**B**) Analysis of the results obtained in A by normalizing the P2 fraction to H (P2/H). (**C**) ACE2 from the apical membranes of control, NB-DNJ-, and dMM-treated cells was treated with endo H followed by Western blot analysis. The blots were taken from different experiments. Tukey’s multiple comparisons test, ** *p* < 0.01, versus Control P2, S.E.M., *n* = 3. Original Western blot images are contained in Appendix A.

**Figure 5 biomolecules-14-00537-f005:**
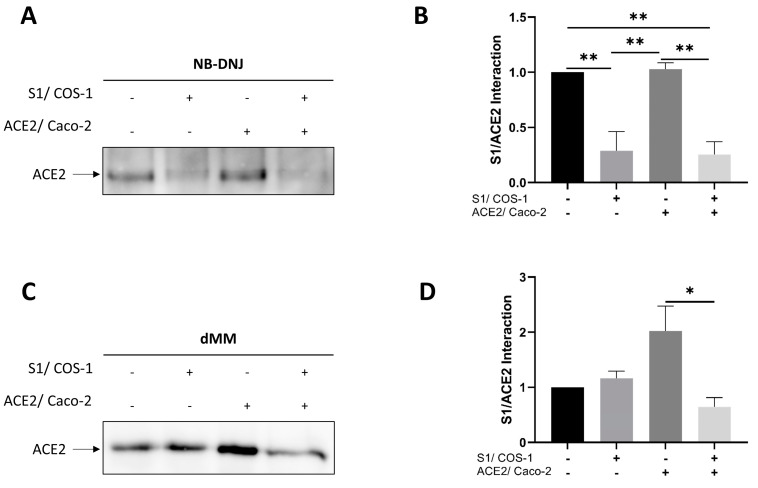
Altered interaction of the modified glycoforms S1 and ACE2. (**A**) Media from control and NB-DNJ-treated COS-1 cells expressing S1 were collected and added to Caco-2 cells for 2 h at 4 °C. The cells were lysed, and S1 was pulled down using PAS beads. The captured S1 proteins were subjected to SDS-PAGE, followed by immunoblotting with anti-ACE2 antibodies to detect interacting ACE2. (**B**) Analysis of the results obtained in A. (**C**) The same procedure used in A was followed for dMM-treated samples. (**D**) Analysis of the results obtained in C. The values represented are normalized to the non-treated COS-1/non-treated Caco-2 (- -) control. Tukey’s multiple comparisons test, * *p* < 0.05, ** *p* < 0.01, versus non-treated COS-1/non-treated Caco-2 (- -), S.E.M., *n* = 3. Original Western blot images are contained in Appendix A.

**Table 1 biomolecules-14-00537-t001:** Primary and secondary antibodies.

Antibody	Concentration (μg/μL)	Dilution	Company	Cat #
Recombinant anti-ACE2 antibody [EPR4435(2)]	0.231	1:5000	Abcam (Cambridge, UK)	ab108252
Protein A (HRP Conjugate)	-	1:1000	Cell Signaling (Danvers, MA, USA)	12291
Goat anti-mouse IgG (H+L) Secondary antibody HRP	0.4	1:5000	Thermo Fisher Scientific (Waltham, MA, USA)	31430
Goat anti-rabbit IgG HRP	0.4	1:5000	Thermo Fisher Scientific	31460

## Data Availability

The data presented in this study are included in the article and Appendix A.

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
