# Peer review of "Glycosylation Modulation Dictates Trafficking and Interaction of SARS-CoV-2 S1 Subunit and ACE2 in Intestinal Epithelial Caco-2 Cells"

_biomolecules, 2024, doi:10.3390/biom14050537_

Round 1

Reviewer 1 Report

Comments and Suggestions for Authors

The present study titled " Glycosylation modulation dictates trafficking and interaction of SARS-CoV-2 S1 subunit and ACE2 in intestinal epithelial cells" by Khoury et al. is particularly interesting. The quality of the writing is good. There are a few comments to improve the manuscript:

 -Researchers should give more detail regarding to modulation of glycosylation in viruses, specifically SARS-CoV-2, and effects their binding to the host. Previous studies in this field should be summarized.

- Please use formal language in the text without using "we.

- Appropriate references should be included in the method section.

-Why did you use EndoH enzyme for the experiment? How about alternative glycosidases? Please explain.

-The quality of all figures may be increased.

-The letters N and O in the N and O glycosylation words should be written in italics.

- The conclusion section should be extended to highlight future remarks, importance, and relevance of their work.

Author Response

We would like to thank the reviewer for the constructive review and comments made to improve our manuscript. Attached are the responses to the specific points raised.

Reviewer 2 Report

Comments and Suggestions for Authors

In this manuscript, the authors investigated the role of glycosylation in S1 and ACE2 by altering protein glycosylation with NB-DNJ and dMM treatments in cells. They found the glycosylation was involved in S1 and ACE2 protein trafficking and interaction. This study may shed light on the potential use of NB-DNJ to treat SARS-CoV-2 infections. 

Major comments:

1.     In figures, authors should show all results of comparison tests done in experiments, including the ones with non-significant P values, which can be noted as “ns”. In text, when authors show the results, they should also include whether the result is significant or not. For example, in Fig.1E, there is no comparison test result of Lec8 versus CHO-K1. Is it significant or not? In the text, the authors mentioned that “the ratio between secreted and intracellular S1 showed a 50% reduction”. Is the reduction result significant or not? In Fig.2B, are the results of comparison test of Lec2 and Lec8 versus CHO-K1 significant or not? The same questions to Fig. 4B and Fig.5D.

2.     In all S1 and ACE2 interaction experiments, authors didn’t show WB of how much S1 in the medium used to incubate with cells expressing. Including these data and the ratio of pull-downed ACE2 to S1, will be helpful to determine the decrease of interaction is caused by secretion defects or glycosylation defects.

3.     About the experiments of Calu3 cells, there is only one sentence in the text. Authors should show the results of this experiment and give some discussions about it comparing with CaCo2 experiments. In SFig.2B, treatment with NB-DNJ increased the ACE2 on apical membrane, while treatment with dMM decreased ACE2 on apical membrane. However in CaCo2 cells, both treatments can cause reductions. Does this suggest same treatment may cause different results in different cell types? What is the conclusion of the experiment in SFig.2D, the interaction of both treatments increased or decreased in this experiment? 

Minor comments:

1.     In all pulldown experiments, PAS beads were used. Do authors know whether glycosylation change affect the interaction between Fc tag and PAS? Did authors try control experiments to test Fc interaction with PAS in different treatments?

2.     In Fig.1, the secreted S1 in Lec2 and Lec8 are smaller than CHO-K1, but the intracellular S1 from three samples have similar size. Can authors give some discussion about these results? Did authors try EndoH or PNGaseF treatment to secreted and intracellular S1 protein in this experiment?

3.     In SFig.2B, the ratio results of NB-DNJ and dMM looked different with the results in SFig.2A. Could authors double-check the results?

Author Response

(The authors gave the same response as above.)
